

# Growth type and relative condition factor as a function of the body shape of deep-water crustaceans in the Colombian Caribbean Sea

Jorge Paramo[1], Alfredo Rodriguez[1,2] and Camilo Quintana[1]

[1] Tropical Fisheries Science and Technology Research Group, University of Magdalena, Santa Marta, Magdalena, Colombia
[2] Doctoral Program in Marine Sciences, University of Magdalena, Santa Marta, Magdalena, Colombia

## ABSTRACT

Length-weight relationships (LWR) and relative condition factor were described for species of deep-water crustaceans caught with bottom trawls in a depth range between 150 and 535 m during August and December of 2009, and March and May of 2010 in the Colombian Caribbean Sea. A linear regression was performed using the logarithmically transformed data to calculate the *a* and *b* coefficients of the LWR for 22 crustacean species corresponding to 13 families and 19 genera and three types of crustaceans (shrimp, crab, lobster). Six crustacean species showed a maximum total length greater than that reported in SeaLifeBase: *Garymunida longipes* (77.00 mm), *Eunephrops bairdii* (220.00 mm), *Metanephrops binghami* (197.46 mm), *Penaeopsis serrata* (149.00 mm), *Polycheles typhlops* (196.27 mm) and *Pleoticus robustus* (240.00 mm). A total of 11 species (50.0%) exhibited isometric growth, five species (22.7%) negative allometric and six species (27.3%) positive allometric. This study shows the first estimates of LWR for 12 species of deep-water crustaceans in the Colombian Caribbean Sea. We demonstrate for the first time that the growth parameters (intercept and slope) of the LWR varying significantly as a function of the body shape of crabs, lobsters and shrimps in deep-water crustaceans.

## INTRODUCTION

Deep-sea crustaceans support important global fisheries and are very important for the conservation of biodiversity as they support a wide variety of species (*Chang et al., 2012*; *Boenish et al., 2022*)); therefore, knowledge of population aspects is relevant for the implementation of management and conservation strategies. New potential deep-water crustacean fishing resources have been identified in the Colombian Caribbean (*Paramo & Saint-Paul, 2012a*; *Paramo & Saint-Paul, 2012b*; *Paramo & Saint-Paul, 2012c*). However, a potential sustainable use of those resources needs an ecosystem approach to fisheries management (EAF) that balance diverse societal objectives, by taking account of biotic, abiotic, and human components of ecosystems and their interactions and applying a

Corresponding author
Jorge Paramo,
jparamo@unimagdalena.edu.co

holistic approach to fisheries management (*Garcia et al., 2003*; *Bianchi, 2008*). Deep-water crustaceans with the highest biomass in the Colombian Caribbean are the deep-water giant red shrimp (*Aristaeomorpha foliacea*, Risso, 1827), the royal red shrimp (*Pleoticus robustus*, Smith, 1885) (*Paramo & Saint-Paul, 2012a*), the pink speckled deep-water shrimp (*Penaeopsis serrata*, Bate, 1881) (*Paramo & Saint-Paul, 2012b*), the deep-water lobster (*Metanephrops binghami*, Boone, 1927) (*Paramo & Saint-Paul, 2012c*), the Squat lobster (*Agononida longipes*, Milne-Edwards, 1880) (*Espitia, Paramo & Wolff, 2019*) and the shrimp (*Pleosionika longipes*, Milne-Edwards, 1881) (*Pérez, Paramo & Wolff, 2019*). However, more scientific information is required about the life cycle, length-weight relationship and population characteristics of deep-water crustaceans, both commercial and non-commercial, before initiating a new commercial fishery. The length-weight relationships (LWR) provide information on the type of growth, the state of the species, habitat conditions and the morphometric characteristics of the species (*Erzini, 1994*; *Gonçalves et al., 1997*; *Morato et al., 2001*; *Froese, 2006*; *Kampouris, Kouroupakis & Batjakas, 2020*; *Falsone et al., 2022*). LWR and condition factor parameters are obtained from length frequency data, which are very useful for estimating biomass and comparing the life history of species between regions (*Erzini, 1994*; *Santos et al., 2002*; *Froese, 2006*). However, LWR parameters may vary between habitats and regions, so accurate estimation of local parameters is essential for comparative studies in stock assessment (*Vaz-dos Santos & Gris, 2016*; *Sousa, Vasconcelos & Riera, 2020*). Additionally, the condition factor based on LWR data is relevant for examining the welfare of populations (*Froese, 2006*; *Koushlesh et al., 2017*; *Jisr et al., 2018*). In data-limited fisheries, often lack sufficient biological information to infer the status of the fish stocks, generally based only on catches, indices, or size classes (*Shephard et al., 2020*). Another issue of limited-data for fisheries management is the insufficient data the information is insufficient to produce quantitative stock assessment (*Dowling et al., 2011*), to determine biological reference points for fisheries management (*Dowling et al., 2011*; *Edwards, 2015*). However, despite their importance, information on LWRs and condition factor are only available for a limited number of species (*Froese, 2006*) and are very scarce in data-limited fisheries from the Colombian Caribbean. In this way, in this work the LWR and the relative condition factor of 22 species of deep-water crustaceans in the Colombian Caribbean Sea were determined, with the purpose of contributing to the knowledge of the biology of deep-water crustaceans.

## MATERIAL AND METHODS

### Study area and sampling

Four surveys were carried out during August and December of 2009, and March and May of 2010, on deep-water ecosystems of the Colombian Caribbean Sea. Biological data of deep-sea crustaceans were collected by trawling at depths 150–535 m. Sampling was carried out on a commercial fishing vessel using a Furuno 1150 echo sounder (28 kHz transducer) (Fig. 1). A trawl net was used with a cod-end mesh size of 44.5 mm from knot to knot. The duration of each trawl on average was 30 min and the distance travelled was calculated using a Global Positioning System (GPS) Garmin MAP 76CSx.

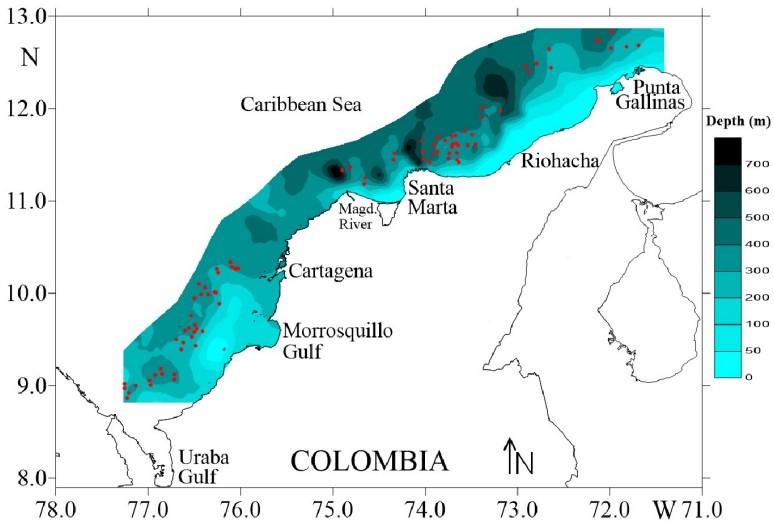

**Figure 1** **Study area in the Colombian Caribbean Sea.** Red circles indicate the sampled stations. Own elaboration by CITEPT-Unimagdalena research group.

The captured individuals were identified to the lowest possible taxonomic level using specialized guides and literature for each taxon (*Carpenter, 2002*). The total length of each individual was measured with a digital calliper with a precision of 0.01 mm, and the total weight was estimated using an analytical balance with an accuracy of 0.01 mg.

The permit, care and use of experimental animals complied with Autoridad Nacional de Licencias Ambientales de Colombia (ANLA), animal welfare laws, guidelines and policies as approved by Universidad del Magdalena reference number 1293-2013.

## Statistical analysis

The information of the parameters of the LWR for each one of the crustacean species was consulted in the SeaLifeBase (https://www.sealifebase.ca/) 06/2023 data base (*Palomares & Pauly, 2023*) and according to information available in the scientific literature. The LWR parameters of the crustacean species were determined by applying the following allometric Eq. (1) (*Keys, 1928*; *Le Cren, 1951*; *Froese, 2006*):

$$W = aL^b \tag{1}$$

where $W$ is the total body weight (g), $L$ the total length (mm), for shrimp and lobster was measured from the tip of the rostrum to the end of the telson, while for crabs the width of the carapace was measured end to end of the lateral spines; $a$ (intercept) y $b$ (slope) are the estimated parameters applying the linear regression model with the log-transformed data (natural logarithm) according to the following Eq. (2):

$$\log W_i = \log a + b \log L_i + \epsilon_i. \tag{2}$$

The corrected back-transformed predicted value of the response variable was calculated by multiplying the back-transformed predicted value by the correction factor (*cf*), where

$RSE$ is the residual standard error and $log_e$ is used to adjust for the base of the natural logarithm (*Ogle, 2016*):

$$cf = e\frac{[log_e RSE]^2}{2} \tag{3}$$

To evaluate the type of isometric growth if $b = 3.0$, negative allometric if $b < 3.0$ and positive allometric if $b > 3.0$, a t-student test was used to determine significant differences from the estimated value of b and its 95% confidence interval (C.I.) (*Zar, 2010*).

To evaluate the influence of body morphology on the growth parameters *a* (intercept) and *b* (slope) of the LWR of the crustacean species, a robust multiple regression model was applied with the data grouped according to the type of crustacean (shrimp, crab, lobster) (*Froese, 2006*).

The relative condition factor (Krel) of the evaluated crustacean was determined according to the following Eq. (3) (*Le Cren, 1951*; *Froese, 2006*):

$$K_{rel} = \frac{W}{aL^b} \tag{4}$$

where $W$ is the observed total body weight (g) of the crustacean specimens and $aL^b$ is the estimated weight from the length-weight relationships. A good growth state of the species was identified when the Krel value $\geq 1.0$, while a species in poor growth conditions when the Krel value $<1.0$ (*Le Cren, 1951*; *Jisr et al., 2018*). A one-sample $t$-test was used to verify significant differences between the Krel and the expected value of Krel $= 1.0$ (*Zar, 2010*). All statistical and graphical analyses were performed in the R 4.2.3 language (*R Core Team, 2023*), using the *modelr*, *FSAmisc*, *moments* and *ggplot2* packages (*Wickham, 2016*; *Komsta & Novomestky, 2022*; *Ogle, 2022*; *Wickham, 2022*).

An analysis of the influence of body morphology on the intercept (a) and slope (b) parameters of the LWR was performed, and linear regression were estimated for three groups of crustaceans: crab, lobster, shrimp. In order to evaluate differences in linear relationships between groups (crab, lobster, shrimp), an analysis of covariance was performed (ANCOVA) once the assumptions of homoscedasticity of the slopes (parallelism) were met with the data transformed into logarithm (*Zar, 2010*).

## RESULTS

A total of 22 crustacean species belonging to 13 families were analyzed, of which the Munididae, Nephropidae and Pandalidae families were the most representative with four, three and three species, respectively (Table 1). Regarding body shape, 10 species showed a shrimp body shape, nine lobsters and three crabs. The most abundant species were *Penaeopsis serrata* (1,714 specimens), *Garymunida longipes* (1099 specimens), *Pleoticus robustus* (930 specimens), *Aristaeomorpha foliacea* (799 specimens), *Glyphocrangon neglecta* (688 specimens), *Metanephrops binghami* (590 specimens), *Achelous spinicarpus* (358 specimens), *Plesionika longipes* (329 specimens), *Glyphocrangon longleyi* (286 specimens) and *Solenocera acuminata* (166 specimens), belonging to families Penaeidae, Munididae, Solenoceridae, Aristeidae, Glyphocrangonidae, Nephropidae, Portunidae, Pandalidae,

Glyphocrangonidae and Solenoceridae, respectively (Table 1). The sizes of the shrimps ranged from 31.79 mm of the pink speckled shrimp (*Penaeopsis serrata*) to 240 mm of the Royal red shrimp (*Pleoticus robustus*), crabs ranged from 13.59 mm of the crab (*Achelous spinicarpus*) to 68.69 mm of the five spine purse crab (*Myropsis quinquespinosa*) and lobsters ranged from 18.33 mm of the Squat lobster (*Antillimunida evermanni*) to 220.00 mm of the Red lobster (*Eunephrops bairdii*).

Species with a maximum total length greater than that reported in SeaLifeBase were *Garymunida longipes* (77.00 mm), *Eunephrops bairdii* (220.00 mm), *Metanephrops binghami* (197.46 mm), *Penaeopsis serrata* (149.00 mm), *Polycheles typhlops* (196.27 mm) and *Pleoticus robustus* (240.00 mm) (Table 1). Linear regressions were significant for all species ($p < 0.05$), with coefficients of determination ($r^2$) between 0.73 and 0.96, except for the lobsters *Garymunida longipes* and *Antillimunida flinti* with the lowest values of 0.56 and 0.68, respectively (Table 2). The intercept of the linear regression (*a*) showed a range of values for shrimps between 6.5326E-08 and 4.4700E-05, for lobsters between 3.5531E-06 and 5.6059E-04 and for crabs between 1.2717E-04 and 5.9070E-04. The slope parameters (*b*) were for shrimps between 2.41 and 4.09, for lobsters between 2.30 and 3.29 and for crabs between 2.95 and 3.31 (Table 2).

According to growth type, 11 species (50.0%) showed isometric growth ($b = 3$), of which two species were shrimps, six were lobsters and three were crabs. However, six species (27.3%) showed positive allometric growth ($b>3$), of which four species were shrimps and two species were lobsters. Nevertheless, five species (22.7%) showed negative allometric growth ($b<3$), of which four were shrimps and one was lobster. It is important to highlight that 17 species evaluated do not have LWR values in SeaLifeBase (https://www.sealifebase.ca/) and the first LWR report for 12 species of deep-water crustaceans in the Colombian Caribbean Sea is shown (Table 2).

The parameters of the LWR linear regression, the intercept log (*a*) and slope (*b*), are highly dependent on the body shape of the crustacean species (Fig. 2). Thus, shrimp species tend to be positive allometric (40.0%), negative allometric (40.0%) and only 20.0% were isometric, while lobster and crab species tend to be isometric, with 66.7% and 100%, respectively, although 22.2% and 11.1% of the lobsters were positive allometric and negative allometric, respectively (Table 2). The results of the ANCOVA revealed that there were significant differences ($p < 0.01$) between the slopes (*b*) of body shape of crabs, lobsters and shrimps in the LWR relationship. When the LWR linear regression was made between the intercept log(a) and the slope b of all crustaceans, the $r^2 = 0.575$ was low. The slope was larger in lobsters ($b = -2.238$) and smaller in crabs ($b = -1.815$) and shrimps ($b = -1.743$) (Fig. 2).

Krel values were greater than 1 in all crustacean species, confirming healthy conditions. The species that showed larger Krel were the shrimp *P. edwardsii* and the lobster *P. typhlops* and *G. longipes* (Fig. 3). Although there are no significant differences (Kruskal-Wallis test, *p*-value = 0.1586) between the Krel, crabs had lowest Krel values on average 1.02, while lobsters and shrimps had 1.03 (Table 2).

Paramo et al. (2024), *PeerJ*, DOI 10.7717/peerj.16583

**Table 1  Descriptive statistics for the 22 deep-sea crustaceans in Colombian Caribbean Sea.**

| Family | Species | Author | N | Body shape | Total length (mm) Mean ± SD (Range) | Total weight (g) Mean ± SD (Range) |
|---|---|---|---|---|---|---|
| Aristeidae | *Aristaeomorpha foliacea* | (Risso, 1827) | 799 | Shrimp | 151.89 ± 28.35 (72.53–225.00) | 22.84 ± 12.23 (2.00–57.20) |
| Calappidae | *Acanthocarpus alexandri* | Stimpson, 1871 | 12 | Crab | 38.83 ± 8.07 (25.54–50.92) | 32.89 ± 20.97 (5.90–76.50) |
| Crangonidae | *Parapontocaris vicina* | (Dardeau & Heard, 1983) | 73 | Shrimp | 78.96 ± 11.51 (54.75–110.12) | 4.99 ± 2.18 (1.50–9.40) |
| Glyphocrangonidae | *Glyphocrangon longleyi* | Schmitt, 1931 | 286 | Shrimp | 82.18 ± 15.38 (54.41–125.00) | 5.76 ± 5.12 (0.70–25.40) |
| | *Glyphocrangon neglecta* | Faxon, 1896 | 688 | Shrimp | 70.09 ± 8.00 (35.69–116.00) | 2.68 ± 2.18 (2.00–57.20) |
| Leucosiidae | *Myropsis quinquespinosa* | Stimpson, 1871 | 45 | Crab | 45.80 ± 10.77 (26.89–68.69) | 36.67 ± 26.86 (5.80–117.90) |
| Munididae | *Antillimunida evermanni* | (Benedict, 1901) | 69 | Lobster | 46.34 ± 7.56 (18.33–73.26) | 4.77 ± 2.98 (0.30–17.90) |
| | *Antillimunida flinti* | (Benedict, 1902) | 12 | Lobster | 48.52 ± 6.51 (36.45–56.41) | 4.63 ± 1.47 (1.90–6.20) |
| | *Babamunida forceps* | (A. Milne Edwards, 1880) | 18 | Lobster | 53.67 ± 9.40 (35.39–67.34) | 5.29 ± 2.64 (1.30–10.50) |
| | *Garymunida longipes* | (A. Milne Edwards, 1880) | 1099 | Lobster | 45.09 ± 6.49 (20.66–**77.00**) | 3.77 ± 1.45 (0.39–11.35) |
| Munidopsidae | *Munidopsis riveroi* | Chace, 1939 | 16 | Lobster | 41.49 ± 7.99 (31.42–56.30) | 3.30 ± 1.99 (1.50–8.40) |
| Nephropidae | *Eunephrops bairdii* | Smith, 1885 | 107 | Lobster | 126.72 ± 34.43 (53.12–**220.00**) | 38.64 ± 37.41 (1.50–218.10) |
| | *Metanephrops binghami* | (Boone, 1927) | 590 | Lobster | 119.56 ± 27.43 (53.65–**197.46**) | 28.65 ± 19.93 (1.70–109.34) |
| | *Nephropsis aculeata* | Smith, 1881 | 113 | Lobster | 70.05 ± 15.53 (43.83–119.16) | 5.32 ± 4.02 (0.90–20.00) |
| Pandalidae | *Heterocarpus ensifer* | A. Milne-Edwards, 1881 | 147 | Shrimp | 100.37 ± 14.81 (63.43–130.30) | 7.60 ± 3.35 (1.50–16.60) |
| | *Plesionika edwardsii* | (Brandt, 1851) | 46 | Shrimp | 109.91 ± 23.22 (60.98–162.00) | 4.25 ± 2.31 (0.70–10.39) |
| | *Plesionika longipes* | (A. Milne-Edwards, 1881) | 329 | Shrimp | 132.62 ± 20.72 (70.87–203.00) | 9.01 ± 4.26 (1.30–36.20) |
| Penaeidae | *Penaeopsis serrata* | Spence Bate, 1881 | 1714 | Shrimp | 104.55 ± 16.66 (31.79–**149.00**) | 5.99 ± 2.69 (0.29–16.30) |
| Polychelidae | *Polycheles typhlops* | Heller, 1862 | 156 | Lobster | 87.94 ± 21.03 (44.00–**196.27**) | 13.63 ± 11.64 (1.00–94.41) |
| Portunidae | *Achelous spinicarpus* | Stimpson, 1871 | 358 | Crab | 24.72 ± 4.83 (13.59–50.19) | 8.55 ± 5.45 (1.30–43.80) |
| Solenoceridae | *Pleoticus robustus* | (Smith, 1885) | 930 | Shrimp | 162.97 ± 30.02 (62.00–**240.00**) | 32.73 ± 18.22 (1.09–96.10) |
| | *Solenocera acuminata* | Pérez Farfante & Bullis, 1973 | 166 | Shrimp | 103.00 ± 18.85 (57.00–199.00) | 12.49 ± 8.13 (1.50–71.90) |

**Notes.**

N, sample size; SD, standard deviation; Bold, maximum total length longer than in SeaLifeBase.

Paramo et al. (2024), *PeerJ*, DOI 10.7717/peerj.16583

**Table 2 Length-weight relationships (LWR) for the 22 deep-water crustaceans in Colombian Caribbean Sea.**

| Family | Species | Body shape | Relationship parameters | | | | | | | Krel Mean ± SD | t-test |
|---|---|---|---|---|---|---|---|---|---|---|---|
| | | | a | 95% C.I. of a | b | 95% C.I. of b | r² | t-student | Growth type | | |
| Aristeidae | *Aristaeomorpha foliacea* | Shrimp | 1.1571E−05 | 7.3505E−06–1.8215E−05 | 2.86 | 2.77–2.95 | 0.83 | 0.00 | Allometric (-) | 1.03 ± 0.28 | 1.00 |
| Calappidae | *Acanthocarpus alexandri* (φ) | Crab | 1.5821E−04 | 1.8009E−05–1.3899E−03 | 3.31 | 2.71–3.90 | 0.94 | 0.28 | Isometric | 1.02 ± 0.18 | 0.61 |
| Crangonidae | *Parapontocaris vicina* (φ)(Δ) | Shrimp | 1.2994E−05 | 3.5077E−06–4.8132E−05 | 2.93 | 2.63–3.23 | 0.84 | 0.63 | Isometric | 1.02 ± 0.19 | 0.78 |
| Glyphocrangonidae | *Glyphocrangon longleyi* | Shrimp | 6.5326E−08 | 3.7290E−08–1.1444E−07 | 4.09 | 3.97–4.22 | 0.93 | 0.00 | Allometric (+) | 1.02 ± 0.20 | 0.95 |
| | *Glyphocrangon neglecta* (φ) | Shrimp | 4.4206E−07 | 2.4300E−07–8.0419E−07 | 3.65 | 3.51–3.79 | 0.79 | 0.00 | Allometric (+) | 1.02 ± 0.32 | 0.98 |
| Leucosiidae | *Myropsis quinquespinosa* (φ)(Δ) | Crab | 1.2717E−04 | 4.3470E−05–3.7200E−04 | 3.23 | 2.95–3.52 | 0.93 | 0.10 | Isometric | 1.03 ± 0.23 | 0.77 |
| Munididae | *Antillimunida evermanni* (φ)(Δ) | Lobster | 5.5929E−05 | 1.7366E−05–1.8012E−04 | 2.93 | 2.63–3.24 | 0.85 | 0.66 | Isometric | 1.02 ± 0.23 | 0.81 |
| | *Antillimunida flinti* (φ)(Δ) | Lobster | 4.1815E−04 | 4.7411E−06–3.6880E−02 | 2.39 | 1.23–3.54 | 0.68 | 0.27 | Isometric | 1.02 ± 0.24 | 0.64 |
| | *Babamunida forceps* (φ)(Δ) | Lobster | 4.0042E−05 | 2.0311E−06–7.8940E−04 | 2.93 | 2.18–3.69 | 0.81 | 0.86 | Isometric | 1.03 ± 0.24 | 0.70 |
| | *Garymunida longipes* (φ) | Lobster | 5.6059E−04 | 3.5499E−04–8.8527E−04 | 2.30 | 2.18–2.42 | 0.56 | 0.00 | Allometric (-) | 1.04 ± 0.29 | 1.00 |
| Munidopsidae | *Munidopsis riveroi* (φ)(Δ) | Lobster | 9.2196E−05 | 1.4943E−05–5.6885E−04 | 2.79 | 2.30–3.28 | 0.91 | 0.37 | Isometric | 1.01 ± 0.15 | 0.62 |
| Nephropidae | *Eunephrops bairdii* (φ)(Δ) | Lobster | 3.5531E−06 | 1.9132E−06–6.5988E−06 | 3.29 | 3.16–3.42 | 0.96 | 0.00 | Allometric (+) | 1.02 ± 0.18 | 0.84 |
| | *Metanephrops binghami* (φ) | Lobster | 3.8607E−06 | 2.7855E−06–5.3509E−06 | 3.27 | 3.20–3.33 | 0.94 | 0.00 | Allometric (+) | 1.02 ± 0.19 | 0.99 |
| | *Nephropsis aculeata* (Δ) | Lobster | 1.0125E−05 | 4.6940E−06–2.1841E−05 | 3.06 | 2.88–3.24 | 0.91 | 0.50 | Isometric | 1.02 ± 0.18 | 0.86 |
| Pandalidae | *Heterocarpus ensifer* (φ)(Δ) | Shrimp | 2.3680E−06 | 1.0187E−06–5.5042E−06 | 3.23 | 3.05–3.42 | 0.89 | 0.01 | Allometric (+) | 1.02 ± 0.17 | 0.86 |
| | *Plesionika edwardsii* (Δ) | Shrimp | 4.4700E−05 | 5.4415E−06–3.6720E−04 | 2.41 | 1.97–2.86 | 0.73 | 0.01 | Allometric (-) | 1.06 ± 0.39 | 0.84 |
| | *Plesionika longipes* (φ)(Δ) | Shrimp | 8.2265E−06 | 4.0646E−06–1.6650E−05 | 2.83 | 2.68–2.97 | 0.82 | 0.02 | Allometric (-) | 1.02 ± 0.23 | 0.97 |
| Penaeidae | *Penaeopsis serrata* (φ) | Shrimp | 3.6517E−05 | 2.7011E−05–4.9370E−05 | 2.57 | 2.50–2.63 | 0.78 | 0.00 | Allometric (-) | 1.03 ± 0.22 | 1.00 |
| Polychelidae | *Polycheles typhlops* (Δ) | Lobster | 4.2898E−05 | 1.4162E−05–1.2994E−04 | 2.79 | 2.54–3.04 | 0.76 | 0.09 | Isometric | 1.06 ± 0.42 | 0.97 |
| Portunidae | *Achelous spinicarpus* (φ) | Crab | 5.9070E−04 | 4.4436E−04–7.8523E−04 | 2.95 | 2.86–3.04 | 0.92 | 0.26 | Isometric | 1.01 ± 0.17 | 0.94 |
| Solenoceridae | *Pleoticus robustus* (φ) | Shrimp | 3.6061E−06 | 2.9410E−06–4.4217E−06 | 3.12 | 3.08–3.16 | 0.96 | 0.00 | Allometric (+) | 1.01 ± 0.13 | 0.97 |
| | *Solenocera acuminata* (φ) | Shrimp | 1.1801E−05 | 5.1777E−06–2.6899E−05 | 2.97 | 2.79–3.15 | 0.87 | 0.72 | Isometric | 1.02 ± 0.20 | 0.91 |

**Notes.**
a, intercept; b, slope; C.I., lower and upper confidence (95%); r², determination coefficient; *t*-student, *p*-value of *t*-student; I, isometric growth; A(-), negative allometric growth; A(+), positive allometric growth; Krel, relative condition factor; *t*-test, *p*-value of one-sample *t*-test; (φ), No available data of LWR in SeaLifeBase; (Δ), First report of the LWR in the Colombian Caribbean Sea.
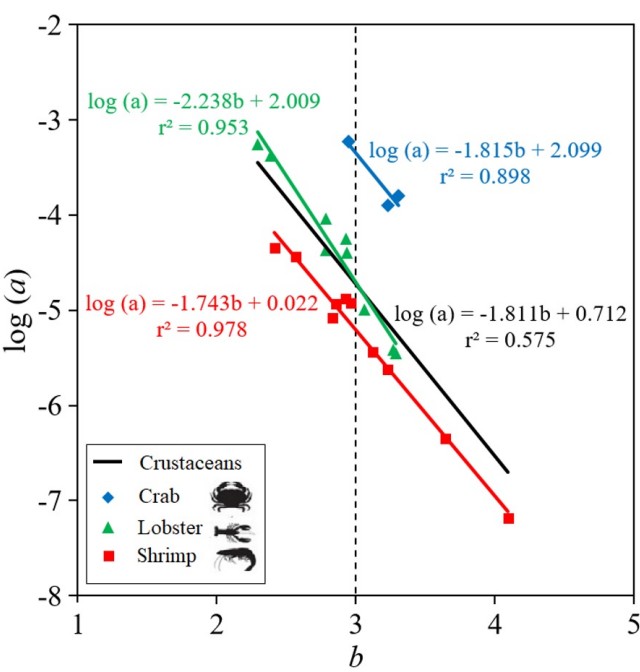

**Figure 2** Scatter plot of mean log *a* over mean *b* for 22 deep-water (three of types of crustaceans: shrimp, crab, lobster) in Colombian Caribbean Sea. All crustaceans: black line; crab: blue rhombus; Lobster: green triangle; shrimp: red square.

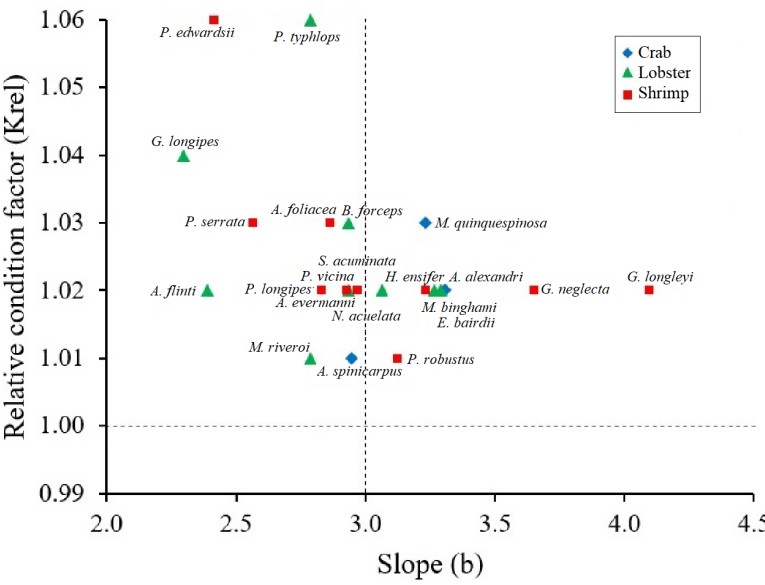

**Figure 3** Scatter of the relationship between the relative condition factor (Krel) and the body condition factor (b) in type of crustaceans (shrimp, crab, lobster). All crustaceans: black line; crab: blue rhombus; Lobster: green triangle; shrimp: red square.

## DISCUSSION

From the point of view of sustainable fisheries management, it is relevant to know the size structure of the populations, especially in these populations of deep-water crustaceans in the Colombian Caribbean that have not been commercially exploited and the growth type and relative condition factor can be considered sustainable biological reference points. In this way, when analyzing the sizes, the demographic parameters of a population can be described in relation to the fishing pressure (*Lizárraga-Cubedo, Pierce & Santos, 2008*). In addition, the patterns of morphometric variation indicate differences in growth, since the shape of the body is the product of ontogeny, that is, structural changes in the development of the organism, which is very important to implement efficient fisheries management measures (*Cadrin, 2005*). Through the determination of the relationships between the size structure of the body morphology of groups of crustaceans, management measures can be implemented for the beginning of a fishery, such as: average size of the catch, selectivity in the fishing gear, type growth (allometric and isometric), relative condition factor, etc. (*Barbosa-Saldaña, Díaz-Jaimes & Uribe-Alcocer, 2012*). Although size is usually measured as a length, weight measurements are required in fisheries to calculate fishing yield, so it is very useful to determine morphometric relationships (*King, 2007*).

Fisheries science seeks the generation of new knowledge to know the aspects that favor the sustainability of marine resources, such as the analysis of length-weight and relative condition factor relationships, which can provide an important insight into the ecology of the species (*Froese, 2006*). *Pauly (1993)* mentions requirements in marine stock assessment where length-weight relationships may be needed, which include: (1) conversion of individual fish length to weight, (2) estimation of mean weight of a given length class, (3) conversion of the length growth equation to a weight growth equation, and (4) morphological comparisons between populations of the same species, or between species. For most fish species, data on length-weight relationships are available from FishBase (http://www.fishbase.org), which is a global fish information system (*Froese & Pauly, 2023*). However, information on crustaceans and cephalopods is available in a similar information system called SeaLifeBase (http://www.sealifebase.org), although with very little information, especially for deep-water crustaceans (*Palomares & Pauly, 2023*).

In the Colombian Caribbean there are very few antecedents of LWR studies in deep-water crustaceans, therefore, it is not possible to determine if there are variations of the parameter b and the type of growth of the species. However, in marine populations, variations in growth are related to factors such as ontogeny, feeding (quantity, quality and size), sex, state of maturity, health, seasonality, habitat, range of sizes and the sample size (*Keys, 1928*; *Le Cren, 1951*; *Safran, 1992*; *Moyle & Cech Jr, 2004*; *Froese, Tsikliras & Stergiou, 2011*; *Correa-Herrera, Jiménez-Segura & Barletta, 2016*).

Four species showed low numbers of specimens, the crab *Acanthocarpus alexandri* ($n = 12$) and the lobsters *Antillimunida flinti* ($n = 12$), *Munidopsis riveroi* ($n = 16$) and *Babamunida forceps* ($n = 18$) (Table 1). However, according to *Froese (2006)* for length-weight relationships a low number of specimens may be acceptable when species are rare, such as these deep-sea crustaceans. A total of 77.3% of the crustacean species evaluated

presented a range of parameter b between 2.5 and 3.5 (*Froese, 2006*), indicating normal growth dimensions (*Bagenal & Tesch, 1978*; *King, 2007*), except in *Antillimunida flinti*, *Garymunida longipes* and *Plesionika edwardsii* with values of $b < 2.5$, while *Glyphocrangon longleyi* and *Glyphocrangon neglecta* with $b > 3.5$, showed a narrow size range, common in values of $b < 2.5$ or $>3.5$ (*Carlander, 1977*; *Froese, 2006*). Additionally, the giant red shrimp *Aristaeomorpha foliacea* showed negative allometric growth off the west coast of Sicily (*Falsone et al., 2022*) and in the Tyrrhenian Sea (*Apostolidis & Stergiou, 2008*), which agrees with our findings. However, it differs from the positive allometric growth reported for this species in the Strait of Sicily (*Ragonese et al., 2004*). The *Plesionika edwardsii* shrimp showed variations in the type of growth with positive allometric in females and negative allometric in males in the Spanish coast of the Western Mediterranean Sea (*García-Rodriguez, Esteban & Pérez-Gil, 2000*; *Company & Sardá, 2000*). In addition, isometric growth was reported in the lobster *Polycheles typhlops* (*Company & Sardá, 2000*), which also coincides with this study.

## CONCLUSIONS

We demonstrate for the first time that the growth parameters (intercept and slope) of the LWR varying significantly as a function of the body shape of crabs, lobsters and shrimps in deep-water crustaceans. Shrimp tend to have allometric growth, positive or negative, but lobster and crabs' growth type is mainly isometric. Regarding the relative condition factor, we found that all the species evaluated had good health conditions, which may be related to the fact that in the Colombian Caribbean the deep-water ecosystem can still be considered pristine and there is currently no fishery; therefore, there is no negative impact from fishing on these deep-water crustacean species (*Paramo & Saint-Paul, 2012a*; *Paramo & Saint-Paul, 2012b*; *Paramo & Saint-Paul, 2012c*). The LWR parameters and the relative condition factor can be applied as simple and inexpensive indicators to implement to assess the well-being of marine populations, considering that a change in the growth conditions of the species may indicate an impact of anthropogenic origin and/or environmental. Finally, although the results of this study were obtained from mixed sexes, they are of great importance for the management of marine resources, since management and conservation regulations are not specific for each sex and can be applied to the entire population (*Falsone et al., 2022*).

## ACKNOWLEDGEMENTS

This study is a contribution of the Tropical Fisheries Science and Technology Research Group (CITEPT) at the Universidad del Magdalena in Colombia. We thank the researchers of the CITEPT Research Group, who collected the data on board the vessel "Adriatic" and analyzed the fish samples in the laboratory.

### Funding

Alfredo Rodriguez was sponsored by the Fondo de Ciencia, Tecnología e Innovación (FCTeI) del Sistema General de Regalías (SGR) and the Doctoral Excellence Scholarship Program Bicentenario del Ministerio de Ciencia, Tecnología e Innovación (Minciencias). The scientific fishery sampling was funded by Autoridad Nacional de Acuicultura y Pesca (AUNAP) and Universidad del Magdalena. The funders had no role in study design, data collection and analysis, decision to publish, or preparation of the manuscript.

### Grant Disclosures

The following grant information was disclosed by the authors:
Fondo de Ciencia, Tecnología e Innovación (FCTeI) del Sistema General de Regalías (SGR).
Doctoral Excellence Scholarship Program Bicentenario del Ministerio de Ciencia, Tecnología e Innovación (Minciencias).
Autoridad Nacional de Acuicultura y Pesca (AUNAP) and Universidad del Magdalena.

### Competing Interests

The authors declare that they have no known competing financial interests or personal relationships that could have appeared to influence the work reported in this article.

### Author Contributions

- Jorge Paramo conceived and designed the experiments, performed the experiments, analyzed the data, prepared figures and/or tables, authored or reviewed drafts of the article, and approved the final draft.
- Alfredo Rodriguez conceived and designed the experiments, performed the experiments, analyzed the data, prepared figures and/or tables, authored or reviewed drafts of the article, and approved the final draft.
- Camilo Quintana conceived and designed the experiments, analyzed the data, prepared figures and/or tables, authored or reviewed drafts of the article, and approved the final draft.

### Field Study Permissions

The following information was supplied relating to field study approvals (*i.e.*, approving body and any reference numbers):
The permit, care and use of experimental animals complied with Autoridad Nacional de Licencias Ambientales de Colombia (ANLA), animal welfare laws, guidelines and policies as approved by Universidad del Magdalena reference number 1293-2013.

### Data Availability

The raw data is available in the Supplemental File.

## Supplemental Information

Supplemental information for this article can be found online at http://dx.doi.org/10.7717/peerj.16583#supplemental-information.

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
