# Peer review of "Growth type and relative condition factor as a function of the body shape of deep-water crustaceans in the Colombian Caribbean Sea"

_PeerJ, doi:10.7717/peerj.16583_

## Round 0.1 · original submission · Minor Revisions

Your manuscript has received comments by two reviewers. Please, follow all suggestions provided.

·

Basic reporting

The manuscript uses clear and professional English, and I find it easy to follow through. However, the Introduction section could benefit from a restructuring and the addition of new literature. The current Introduction section is too short, and all points are clustered together. Kindly, in a few paragraphs, introduce the importance of deep-sea crustaceans, the sampling site, LWR, condition factor etc. This will provide a sufficient introduction to the readers.

The tables and figures are suitable. However, since some species have a small sample size as noted in Table 1, are the LWRs derived from these potentially skewed data still valid (Table 2)?

Experimental design

The framework of this study is sound and valid. Methods were thoroughly described and reproducible. The quotations used to measure different parameters for different crustaceans are correct. Samples were collected during surveys using the trawling method. However, I could not find the permit for the surveys. Could the authors kindly supply us with the approval number for the four survey samplings? The ethical statement at Line 252 only concerns the ethical use of animals but not permit.

Validity of the findings

The results are well described. However, the Discussion could benefit from a more in-depth analysis. The authors could expand on the final paragraph of the Discussion, and look specifically into the patterns portrayed. For example, was the growth allometry pattern genus or family-specific? or species-specific? Does body size play a role in growth allometry patterns - larger-sized crustaceans often show similar growth patterns? Any reported LWR studies on some of the investigated species?

Additional comments

Additional comments:
Line 222: What are these "Three species"?

·

Basic reporting

No comments. I find this manuscript being well-written, text is clear and unambiguous. References are sufficient and include the main recent reviews, and the structure is the usual for this kind of contributions.

Experimental design

No comments. The submission identifies as a gap the lack of basic knowledge on the length-weight relationships of crustaceans in Colombian waters, and contributes to filling this gap with an extensive study of a large number of species, using appropiate methods.

Validity of the findings

No comments. The manuscript provides lenght-weight relationships for a large number of species, which are valuable data to update the SeaLifeBase. Primary data are extensive and thus not provided, but the authors stated that these are available upon request.

Additional comments

Regarding the statistical analysis, authors used the logarithmic transformation of the data to fit the linear regression model; but they do not state whether they used log(10) or natural logarithms in Eq. 2. From Eq. 3 it seems that they used log(10), then followed Ogle (2016) in applying this correction factor to “adjust for the base of the logarithm used”.

I find this somewhat confusing, as it gives the impression that the need for correction merges from the selected base. If natural logarithms were used instead, would this correction be required? If not, why not using natural logarithms in the first place? On the other hand, I understand that a correction would be required after log(e) transformation, as the back-transformed mean value from the log scale is equal to the geometric mean of the values on the original scale, and the geometric mean is less than the arithmetic mean. Is this correction also considered by Eq. 3?

Regarding their conclusions, authors claim that they “demonstrate for the first time the strong correlations of the growth parameters (intercept and slope) of the LWR varying significantly as a function of the body shape of crabs, lobsters and shrimps in deep-water crustaceans”. Though they highlight this as a relevant aspect of their contribution (even in the abstract), they do not discuss the exact nature and the extent of this claim. As the authors mention elsewhere, LWR vary widely among different taxa, between different populations of the same species, and even within the same population at different times. Moreover, differences shown in Figure 2 between crabs, lobsters and shrimps are at least partially determined by the differences in the methodology (crabs length was measured as width of the carapace). In brief, these differences are to be expected and therefore I do not get the relevance of this issue. I feel, however, that I am missing the point; please elaborate it further in the discussion section.

Some minor comments are as follows:

Line 49: "Pérez et al (2020)", should be Pérez et al. (2019).

Lines 52-56: “The length-weight relationships (LWR) provide information…, mainly in deep-water crustaceans with potential for a new fishery in the Colombian Caribbean (Paramo & Saint-Paul, 2012a,b,c)”. Consider rephrasing (LWR provide this information in all cases, not mainly in these).

Lines 62-64: “However, despite their importance, information on LWRs and condition factor are only available for a limited number of species (Kulbicki et al., 2005; Froese, 2006)”. The Kulbicki et al. (2005) reference regards lagoon fishes in New Caledonia and does not seem pertinent as a review on this topic.

Lines 80-81: “The captured individuals were identified to the lowest possible taxonomic level using specialized guides and literature for each taxon (Carpenter, 2002a,b)”. Carpenter 2002b refers to bony fishes, not crustaceans.

---

## Round 0.2 · accepted · Accept

You addressed the suggested changes provided by reviewers well.